# The Ameliorative Effect of COST on Diet-Induced Lipid Metabolism Disorders by Regulating Intestinal Microbiota

**DOI:** 10.3390/md20070444

**Published:** 2022-07-07

**Authors:** Huimin You, Xiaoyi Deng, Yan Bai, Jincan He, Hua Cao, Qishi Che, Jiao Guo, Zhengquan Su

**Affiliations:** 1Guangdong Engineering Research Center of Natural Products and New Drugs, Guangdong Provincial University Engineering Technology Research Center of Natural Products and Drugs, Guangdong Pharmaceutical University, Guangzhou 510006, China; yhm98815@163.com (H.Y.); dxy1500718043@163.com (X.D.); 2Guangdong Metabolic Disease Research Center of Integrated Chinese and Western Medicine, Key Laboratory of Glucolipid Metabolic Disorder, Ministry of Education of China, Guangdong TCM Key Laboratory for Metabolic Diseases, Guangdong Pharmaceutical University, Guangzhou 510006, China; 3School of Public Health, Guangdong Pharmaceutical University, Guangzhou 510310, China; angell_bai@163.com (Y.B.); hejincan300@163.com (J.H.); 4School of Chemistry and Chemical Engineering, Guangdong Pharmaceutical University, Zhongshan 528458, China; caohua@gdpu.edu.cn; 5Guangzhou Rainhome Pharm & Tech Co., Ltd., Science City, Guangzhou 510663, China; cheqishi@rhkj.com.cn

**Keywords:** chitosan oligosaccharides, gut microbes, disorders lipid metabolism, diet

## Abstract

Background: Chitosan oligosaccharides, with an average molecular weight ≤ 1000 Da (COST), is a natural marine product that has the potential to improve intestinal microflora and resist lipid metabolism disorders. Methods: First, by establishing a mice model of lipid metabolism disorder induced by a high fat and high sugar diet, it is proven that COST can reduce lipid metabolism disorder, which may play a role in regulating intestinal microorganisms. Then, the key role of COST in the treatment of intestinal microorganisms is further confirmed through the method of COST-treated feces and fecal bacteria transplantation. Conclusions: intestinal microbiota plays a key role in COST inhibition of lipid metabolism disorder induced by a high fat and high sugar diet. In particular, COST may play a central regulatory role in microbiota, including *Bacteroides*, *Akkermansia*, and *Desulfovibrio*. Taken together, our work suggests that COST may improve the composition of gut microbes, increase the abundance of beneficial bacteria, improve lipid metabolism disorders, and inhibit the development of metabolic disorders.

## 1. Introduction

In recent years, due to the gradual improvement of living standards, residents’ intake of high fat and high sugar food has increased significantly. In recent years, many studies have shown that a high fat and high sugar diet leads to hyperlipidemia, atherosclerosis, diabetes, and lipid metabolism disorders [1,2]. Despite tremendous efforts to combat diet-related lipid metabolism disorders, progress has been slow over the past decade in understanding the causes and mechanisms that cause these problems, which may lead to the identification of possible treatments [3]. Recently, growing evidence has shown that intestinal microbiome dysregulation is closely related to the host’s metabolism. The human gut microbiome is a complex ecosystem populated by thousands of species of microorganisms. It varies from person to person depending on the host’s genotype and environmental factors such as diet and antibiotics [4]. In addition, the gut microbiome is involved in the etiology of diet-related complications associated with lipid metabolism disorders. Recent studies have shown that controlling dietary choices has a significant impact on outcomes and interpretations related to dietary effects on gut microbiota [5]. In the last 15 years, the gut microbiome has become an important research focus as a regulator of the host’s energy metabolism and substrate metabolism. Abnormalities in the composition and function of the gut microbiome can lead to disturbances in energy and substrate metabolism, including effects on adipose tissue, liver, and muscle metabolism [6,7]. Recent studies have shown that a high fat and high sugar diet (HFHSD) can also cause intestinal microbe imbalance, and an unhealthy diet can drive gastrointestinal tumors by changing the composition of intestinal microbes. Therefore, the intervention of intestinal microbes to improve lipid metabolism disorders has become one of the current effective strategies [7,8].

Chitosan is a natural product that comes from the ocean [9,10]. It is a typical product of synthetic advanced chemical reactions and the degradation of shrimp and crab shells. Cluster analysis showed that chitosan had reasonable weight and lipid reduction activity. However, its poor water solubility and adsorption greatly restrict its development and utilization. A chitosan product, chitosan oligosaccharides (COS) from a wide range of sources, has the characteristics of low cost, high water solubility, high bioavailability, and biological activities [11,12]. Therefore, COS is widely used in the biomedical and health food fields. Consistent with current reports, chitosan has good effects on elevated blood glucose, elevated lipids, inflammation, tumors, and other diseases. In this study, chitosan oligosaccharides with an average molecular weight of 1000 Dalton (COST) were used based on previous studies. Therefore, we speculated that COST has a good regulatory effect on diet-induced lipid metabolic disorder and may play a role by regulating the intestinal microflora structure. Therefore, we established a lipid metabolic disorder model induced by a high fat and high sugar diet and adopted the pseudogerm-free mice mouse model and fecal microbiota transplantation (FMT) method. Core microorganisms that may play a key role after COST treatment were identified. Our study will provide new insights into the potential mechanisms of COST-mediated resistance to diet-induced lipid metabolism disorders.

## 2. Results

### 2.1. Effect of COST Intervention on Serum Lipid Levels and Liver Lipid Accumulation in Mice

Mice fed a normal diet (CTRL group) weighed less than those fed an HFHSD (MOD group), and mice treated with high-dose COST (COST-H group) also had lower body weights than those in the MOD group. However, no significant difference was found, and the body weight of the mice in the medium-dose COST treatment group (COST-M group) and low-dose COST treatment group (COST-L group) was similar to that of the MOD group (Appendix A). Compared with CTRL mice, the serum levels of total cholesterol (TC) (*p* < 0.0001), total triglyceride (TG) (*p* < 0.0001), low-density lipoprotein cholesterol (LDL-C) (*p* < 0.0001), and nonesterified fatty acids (NEFA) (*p* < 0.0001) were significantly increased in MOD mice, while the serum HDL-C level was decreased (*p* < 0.0001). The plasma levels of TG, TC, and LDL-C decreased in the high-dose and medium-dose COST groups (*p* < 0.01) compared with the MOD group (Figure 1a,b,d; Appendix A). NEFA levels in the high-dose, medium-dose, and low-dose COST groups were significantly lower than those in the MOD group (*p* < 0.0001) (Figure 1e, Appendix A). The serum lipoprotein cholesterol (HDL-C) levels in the COST-H and COST-M groups were similar to those in the CTRL group and showed an upward trend compared with the MOD group (Figure 1c; Appendix A). Studies have found that lipid metabolism disorders can affect abnormal hyperglycemia and possibly insulin resistance (IR) [13,14]. In addition, the serum glucose concentration in the MOD group was significantly higher than that in the CTRL group, and the serum glucose concentration in the high-dose COST group was lower than that in the MOD group (*p* < 0.0001), and according to the result of the oral glucose tolerance test (OGTT) (Figure 1f,g; Appendix A), all indicating that COST has a hypoglycemic effect and can enhance the glucose tolerance of mice. In addition, the liver morphology of mice after COST intervention was observed. As shown in Figure 1h, the liver cells of mice in the CTRL group were arranged in an orderly manner, with a clear nuclear structure and cell boundaries and orderly cells without fatty degeneration and necrosis. In the MOD group, liver cells were disordered, the boundary between cells was blurred, and fat vacuoles of different sizes appeared inside the cells, leading to steatosis. The hepatic cell structure and hepatic cord arrangement of the COST-H and COST-M groups were basically close to normal, and basically, no fat vacuoles were found. The liver pathology of the COST-L group was significantly improved, but a few liver vacuoles were still found. This indicates that COST can be used to intervene in the lipid metabolism disorder caused by HFHSD. In addition, the liver histopathology score was scored according to the NASH CRN scoring system [15,16]; the NAFLD activity score (NAS) was calculated from the sum of individual scores for steatosis, inflammation, and ballooning degeneration. After COST treatment (Figure 1i,j), the NAS score was also significantly reduced, especially for the steatosis score (*p* < 0.01).

### 2.2. Effect of COST Intervention on Intestinal Microflora Structure in Mice

According to the above activity results, COST can reduce lipid accumulation and has the potential to resist lipid metabolism disorders, and one of the key features of lipid metabolism disorders is the influence of gut microbes [17,18,19,20]. Therefore, we wanted to further study the therapeutic effect of COST on lipid metabolism disorders by regulating intestinal flora. In addition, according to the analysis results in the previous part, COST-H has the best activity in improving lipid metabolism disorders, so we next selected the fecal samples of the COST-H dose group for the following intestinal flora analysis. For these data, “COST group” refers to the “COST-H group”. According to the OTU results obtained by clustering (Figure 2a), the three groups of mice were compared to evaluate the differences in intestinal flora composition. The distribution results showed that 627 OTUs were obtained, including 504 OTUs in the CTRL group, 522 OTUs in the MOD group, and 463 OTUs in the COST group. HFHSD induced changes in the OTU number of mice with lipid metabolism disorders, and COST intervention had a certain impact on the changes in the OTU number of mice with lipid metabolism disorders. The phylum species were analyzed based on OTUs. Next, the relative abundance of bacteria was further evaluated at the phylum and genus levels. At the phylum level, *Bacteroidetes*, *Firmicutes,* and *Proteobacteria* were confirmed to be the most abundant bacteria in the feces of the three groups of mice, and the relative abundance of the three phyla could reach 90%; thus, they were the absolute dominant bacteria in the intestinal flora (Figure 2b). Compared with the MOD group, the abundance of *Firmicutes* and *Proteobacteria* in the CTRL group was significantly decreased (*p* < 0.001, *p* < 0.01), while the abundance of *Bacteroidetes* was increased (*p* < 0.001). Compared with the MOD group, the COST group was able to upregulate the abundance of *Bacteroidetes* (*p* < 0.05) and downregulate the abundance of *Proteobacteria* (*p* < 0.05) (Figure 2d). Similarly, the microbiota structure of each group was different at the genus level (Figure 2e,f). The *Bacteroides*, *Lachnospiraceae,* and *Akkermansia* genera were significantly upregulated by the intervention of the CTRL group and COST group compared with the MOD group (*p* <0.05). *Desulfovibrio* bacteria was downregulated. This indicates that the intervention of the COST group can lead to a certain impact on *Lachnospiraceae*, *Bacteroides*, *Desulfovibrio,* and *Akkermansia* bacteria in the microorganism. The *Firmicutes*/*Bacteroidetes* (F/B) ratio has been recognized in the scientific literature as a marker of lipid metabolism disorders [21,22]. The intestinal F/B ratio in the CTRL group was the lowest, and the F/B ratio in the MOD group was significantly higher than that in the CTRL group (*p* < 0.0001) and COST group (*p* < 0.05) (Figure 2c). This indicates that COST reverses lipid metabolism disorders through the flora. We also analyzed the Observed species, Chao1, ACE, and Simpson index (Figure 2g), and the results showed that HFHSD increased the diversity of gut microbes in mice, and the diversity was downregulated in the CTRL and COST groups compared with the MOD group, suggesting that COST treatment partially reverses the increase in gut microbiota diversity and richness caused by HFHSD feeding. The results of the principal coordinate analysis (PCoA) based on unweighted UniFrac metrics (Figure 2h) showed significant differences in microbiome structure among the three groups. Changes in the composition and abundance of gut microbiota were accompanied by changes in function. Tax4Fun was used to predict the metabolic function of intestinal microorganisms in this study. We were interested in functions related to metabolic pathways; thus, we further analyzed them (Figure 2i) and noted nine pathways in KEGG metabolism. The results showed that carbohydrates metabolism and glycan biosynthesis and metabolism of the COST group were slightly higher than those of the MOD group.

### 2.3. Effects of FMT on Body Weight and Liver in Mice

Next, the fecal bacteria transplantation experiment was carried out on pseudogerm-free mice that were successfully modeled. The specific experimental method is shown in Figure 3a. During the FMT experiment, the weekly body weight of C57BL/6J mice in each group was recorded (Figure 3b, Appendix A). Macroscopically, compared with other FMT groups, the Model group gained more weight, and there were significant differences between the Model group and the other four groups. The fecal transplant from the MOD group mice with lipid metabolism disorder can promote lipid metabolism disorder. The results of the COST-H-FMT, COST-M-FMT, and COST-L-FMT groups showed that the high-dose group, in particular, could maintain the initial body weight of mice well and could effectively resist the effect of promoting lipid metabolism disorder caused by the high fat and high sugar diet compared with the Model group (*p* < 0.001), and all could inhibit the metabolic disorders caused by the HFHSD diet. No significant difference was found in the final body weight of FMT mice treated with three doses of COST; therefore, no significant dose-dependent effect of COST transplantation intervention was found. The liver weight of the Model group was compared (Figure 3c, Appendix A), and the liver weight of the Model group was significantly higher than that of the COST-H-FMT, COST-M-FMT, and COST-L-FMT groups (*p* < 0.05) Next, our observation of pathological tissue sections in each group showed (Figure 3f) no obvious liver lipid deposition in the STD, Control, COST-H-FMT, COST-M-FMT, and COST-L-FMT groups. However, liver lipid accumulation in the Model group was obvious. After transplantation of feces of mice receiving COST intervention (Figure 3d,e), the NAS score was also significantly reduced, especially for the steatosis score (*p* < 0.001). These results suggest that transplantation of feces from donor mice with metabolic disorders can increase the liver weight and the risk of liver steatosis in recipient mice, and transplantation of feces of mice receiving COST intervention can effectively improve this situation.

### 2.4. Effects of FMT on Adipose Tissue in Mice

The main function of white adipose tissue (WAT) is to store excess fat in the body, which is a common mechanism and manifestation of lipid metabolism. The main function of brown adipose tissue (BAT) is to break down lipid molecules and convert them into CO_2_, H_2_O, and heat energy, which helps increase the body’s metabolism and WAT consumption and is converted into beige fat cells [23,24]. Therefore, we compared the adipose tissues of mice in each group. For BAT (Figure 4a, Appendix A), the content of BAT in the STD group was the highest, while that in the Model group was the lowest, but the difference was not significant. On the other hand, WAT analysis of different parts (Figure 4b–d, Appendix A) showed that fecal bacteria transplantation of different donor groups in the FMT experiment had a considerable impact on the WAT of mice. The WAT content of the standard diet and transplanted intestinal flora in the STD and Control groups were significantly lower than that in the Model group (*p* < 0.05). Similarly, the WAT contents in the COST-H-FMT COST-M-FMT, and COST-L-FMT trans-plantation groups were significantly lower than those in the MOD transplantation group (*p* < 0.001). This result indicated that the intestinal flora of mice fed a normal diet and mice after COST intervention could effectively resist the accumulation of WAT. Next, we also observed pathological sections of perirenal adipose tissue (Figure 4e) and brown adipose tissue (Figure 4f) and observed that the fat cells of the STD group and Control group had normal structures, orderly arrangements, small cell volumes, and a large number of fat cells in the same field of vision. In contrast, the WAT cells in the Model group were uneven in size, with a small number of cells and more obvious cell proliferation and amplification. The sections of the COST-H-FMT and COST-M-FMT groups were similar to those of the STD and Control groups. Compared with the Model group, the adipocytes were significantly smaller with uniform size, smaller volume, and better lesions, but COST-L-FMT showed no significant improvement effect. These results indicate that the treatment of intestinal flora with COST can indirectly inhibit the fat cell hypertrophy caused by the HFHSD, regulate the growth and expansion of fat cells, reduce fat accumulation, and improve the symptoms of abnormal lipid metabolism.

### 2.5. Effects of FMT on Serum and Liver Biochemical Indexes in Mice

Compared with the Model group, the serum TC, TG, and LDL-C in the Control group, COST-H-FMT, COST-M-FMT, and COST-L-FMT groups were significantly decreased in the FMT experiment (Figure 5a–c, Appendix A) (*p* <0.01). Compared with the Model group, the COST-FMT-H, COST-M-FMT, and COST-L-FMT groups also had significantly increased serum HDL-C (Figure 5d, Appendix A) (*p* < 0.01). Disorders of lipid metabolism, such as obesity and nonalcoholic fatty liver disease, can lead to abnormal elevation of serum aspartate aminotransferase (AST) and alanine aminotransferase (ALT). The serum AST and ALT concentrations in the Model group (Figure 5e,f, Appendix A) were significantly higher than those in the STD group (*p* <0.001). Compared with the Model group, the serum ALT concentrations in the Control group, COST-H-FMT, COST-M-FMT, and COST-L-FMT groups were significantly decreased (*p* < 0.001). Similarly, the serum AST content was significantly decreased in the Control group, COST-H-FMT, COST-M-FMT, and COST-L-FMT groups (*p* < 0.001). The levels of glucose in the Model group were also compared (Figure 5g, Appendix A), and the glucose in the Model group was significantly higher than that in the STD group (*p* < 0.001). In contrast, transplantation of feces after COST intervention showed a good effect on adjuvant glucose regulation. The effect of COST on intestinal flora may be to reduce dietary polysaccharides in high fat and high sugar diets, but this effect is not dose-dependent (*p* < 0.001). In addition, according to the results of OGTT (Figure 5h), the blood glucose in the model group reached an abnormally high level within 15 min after oral glucose, failing to return to the initial level after 120 min. These results indicated that the feces of transplanted metabolic disorder mice did not have a benign regulatory effect on blood glucose and was associated with a higher risk of elevated blood sugar caused by lipid metabolism disorder. In contrast, COST-treated bacteria were more effective in helping maintain dynamic blood glucose balance in vivo after transplantation. NEFA and total bile acids (TBA) [25,26] are also closely related to microbial and metabolic disorders. Elevated levels of TBA are also associated with cholestasis and hepatic lipid metabolism disorders [27,28]. The Model group showed high levels of NEFA and TBA, significantly different from the standard diet STD group (*p* < 0.001). Compared with the Model group, the serum concentrations of NEFA and TBA in the Control group, COST-H-FMT, COST-M-FMT, and COST-L-FMT groups were significantly reduced (Figure 5i,j, Appendix A) (*p* < 0.05). COST-treated bacteria reduced dietary fat intake and serum fatty acid and bile acid levels in the high fat and high sugar diet group. In general, the lipid metabolite levels of the Model group of the mouse microflora of the transplanted MOD group showed an opposite trend to that of the healthy STD group; meanwhile, the lipid metabolite levels of the Control group, COST-H-FMT, COST-M-FMT, and COST-L-FMT groups were similar to that of the STD group, suggesting that the intestinal microbiota metabolites were affected by COST intervention. In addition, these results indicate that after COST intervention, the microflora changes in a direction that is beneficial to the body, thus improving the serum metabolite abnormalities caused by a high fat, high sugar diet. Combined with previous analyses of serum lipid levels and metabolic levels, the next step is to assess lipid levels in the liver and TBA related to gut microbial metabolism. The results (Figure 5k–n, Appendix A) showed that the levels of TG, TC, LDL-C, and TBA in the liver of the Model group were significantly higher than those of the STD group (all *p* < 0.01), indicating that feces transplantation from metabolic disorder Model mice could not alleviate the abnormal levels of lipid metabolites in the liver. It may also aggravate the development of obesity and nonalcoholic fatty liver disease and related metabolic diseases. In contrast, the microflora of COST-treated mice effectively alleviated the increase in liver lipid and TBA levels, especially in the COST-H-FMT and COST-M-FMT groups (*p* < 0.05). The regulation of TC, TG, LDL-C, and TBA was even slightly better than that of the Control group transplanted with feces from the CTRL group. COST treatment is beneficial for regulating the balance of liver lipid metabolism, especially in improving lipid metabolism disorders

### 2.6. Structural Characteristics of Intestinal Flora in Mice Treated with FMT

Next, we analyzed the composition of the mice’s gut microbiome. In addition, according to the analysis results in the previous part, COST-H-FMT has the best activity in improving lipid metabolism disorders, so we next selected the fecal samples of the COST-H-FMT dose group for the following analysis. For these data, “COST-H-FMT group” refers to the “COST-FMT group”. According to the OTU results obtained by clustering (Figure 6a), the four groups of mice were compared to evaluate the differences in intestinal flora composition. The distribution results showed that the STD group maintained the standard diet with the largest variety of bacteria, while the number of OTUs in the Model, Control, and COST-FMT groups was less than that in the STD group. The number of OTUs in the COST-FMT group was larger than that in the Model and even slightly better than that in the Control group. According to the OTU results, 10 species with the most abundant phylum and genus classification levels were selected from the 4 groups (Figure 6b,d). At the phylum level, *Bacteroidetes*, *Firmicutes,* and *Proteobacteria* were the most abundant bacteria in the feces of the four groups of mice, and the relative abundance of the three phyla was up to 95%; thus, they were the absolute dominant bacteria in the intestinal flora. Compared with the Model group, *Bacteroidetes* was upregulated in the STD group (*p* < 0.0001), Control group, and COST-FMT group. However, compared with the Model group, the STD group and Control group exhibited downregulated *Proteobacteria* (*p* < 0.05), while COST-FMT showed no significant difference compared with the Model group. Compared with the Model group, the level of *Firmicutes* in the STD group (*p* < 0.05), Control group, and COST-FMT group was downregulated (*p* < 0.05). We also compared the F/B values of the four groups of mice (Figure 6c). Compared with the Model group, the F/B values of the STD group (*p* < 0.0001), Control group, and COST-FMT group (*p* < 0.0001) were significantly decreased, suggesting that fecal transplantation of Control and COST group mice could be resistant to metabolic disorders caused by the HFHSD diet. Second, the microbiome structure of each group was different at the genus level (Figure 6e,f). We analyzed the intestinal microbes of the mice in each group at the genus level; compared with the Model group, *Bacteroides*, *Lachnospiraceae*, *Lactobacillus,* and *Oscillospira* were upregulated, and *Desulfovibrio* was downregulated in the STD group, Control group, and COST-FMT group. This indicates that COST-FMT intervention can cause some effect on *Bacteroides*, *Lactobacillus*, *Lachnospiraceae*, *Oscillospira,* and *Desulfovibrio* bacteria in fecal flora. The author also analyzed the species composition heatmaps at the genus level of the intestinal contents of four groups of mice (Figure 6g) and found that COST has a good regulatory effect on *Clostridium*, *Dehalobacterium*, *Adlercreutzia*, *Akkermansia,* and *Streptococcus*. In addition, we analyzed the Observed species, Shannon, Simpson, and Chao1 indexes (Figure 6h), and the results showed that the feces of HFHSD-transplanted mice (MOD group) could not reverse the decrease in bacterial diversity and richness caused by antibiotic-treatment, while compared with the Model group, the STD group, Control group, and COST-FMT group were upregulated to a certain extent. Finally, the results of the principal coordinate analysis (PCoA) based on unweighted UniFrac metrics (Figure 6i) showed significant differences in microbiome structure among the four groups. A longer distance was observed in the STD group, which was significantly different from the other groups, suggesting that the effect of the standard diet and the HFHSD diet on change was very significant in the gut microbiota. The three groups of FMT mice fed high fat and high sugar diets were relatively close, and the Control group and the Model group partially intersected, indicating that the sample groups had similar structures. In the Control group, the HFHSD during the FMT experiment changed the intestinal microbiota structure relative to the standard diet of the original donor mice. In contrast, COST-treated mice were significantly different from the other two groups, and COST specifically regulated the HFHSD-induced colony structure to a certain extent.

### 2.7. Effects of COST on Liver Lipid Metabolism-Related Gene Expression in FMT Mice

The above activity and intestinal microbe analysis indicated that COST might ameliorate HFHSD-induced lipid metabolism disorders by modulating gut microbes. Next, this study investigated the expression of SREBP, FAS, and PPARγ genes related to lipid metabolism in liver tissues of mice in the STD, Control, Model, and COST-FMT groups (Figure 7a–c). Compared with the Model group, the expression of the SREBP1 and FAS genes in the livers of the STD group mice fed an ordinary diet, the Control group given normal mouse feces, and the mice with feces transplanted after COST intervention significantly decreased (*p* < 0.001). In addition, compared with the Model group, the expression of PPARγ was decreased in the STD and Control groups (*p* < 0.01) and in the COST-FMT group, but the difference was not significant. These structures suggest that the flora treated with COST has an inhibitory effect on increases in hepatic adipose tissue. In addition, PPARα plays a key role in the transcriptional regulation of fatty acid β-oxidation genes. In this experiment, the gene expression of PPARα in the liver of each group of mice was quantified (Figure 7d). The results showed that compared with the Model group, the expression of PPARα was significantly increased in the STD and COST-FMT groups (*p* < 0.001, *p* < 0.05); the expression of PPARα was also increased in the Control group, but the difference was not significant. These results suggest that transplantation in the Control group and COST-FMT group can promote the transformation between peripheral cholesterol and bile acids and promote liver fatty acid catabolic metabolism. Cholesterol α-enhancing enzyme (CYP7A1) is a major rate-limiting enzyme that can catalyze the decomposition of cholesterol into bile acids, and its expression can be activated by a series of active substances to maintain the normal level of cholesterol in the body. In this experiment, the expression of the CYP7A1 gene in the livers of mice in each group was detected (Figure 7e). Compared with the Model group, the expression of the STD group, Control group, and COST-FMT group was significantly decreased (*p* < 0.001). These results showed that the transplantation of bacterial flora also played a beneficial role in lipid metabolism disorders.

## 3. Discussion

In this study, we first studied the role of COST in diet-induced lipid metabolism disorder and the reactivity of intestinal microorganisms to it. COST significantly decreased lipid levels in serum, increased and enhanced the glucose tolerance of mice, and alleviated the accumulation of liver lipids, especially in the COST-H group. The intestinal flora is an important regulator of host metabolic homeostasis and energy balance. In particular, metabolites produced by intestinal flora play a crucial role in disorders of lipid metabolism such as obesity and nonalcoholic fatty liver disease. COST increased the diversity of intestinal microflora, increased the relative abundance of *Bacteroidetes*, and decreased the proportion of *Bacteroidetes* and *Firmicutes* in diet-induced lipid metabolism disorder mice. At the genus level, compared with the MOD group, COST reduced the relative abundance of the *Desulfovibrio* genera and increased the relative abundance of *Bacteroides*, *Lachnospiraceae,* and *Akkermansia*. Among them, *Desulfovibrio* was pathogenic bacteria and closely related to the occurrence of obesity and hyperlipidemia [29,30]. In a large number of studies, the *Bacteroides* strain has also been proven to regulate the redox level in the intestinal tract, creating many favorable conditions for the host [31]. *Lachnospiraceae* has also been proven to play a beneficial role in reducing obesity [32]. *Akkermansia* is a member of *Verrucomicrobia*, and it is conducive to the stability of blood sugar. It degrades mucin and reverses fat gain caused by an HFHSD, metabolic toxemia, and insulin resistance [33,34]. This indicates that COST can inhibit the disorder of lipid metabolism caused by diet by interfering with the regulation of intestinal flora. Therefore, further analysis is needed to determine the intestinal flora and how it can promote the ability of COST to inhibit lipid metabolism disorders. To further study the effect of intestinal flora on dietary anti-lipid metabolism disorder in COST, we used an antibiotic mixture to remove most intestinal flora in mice (Appendix A) and analyzed the intestinal flora in their feces. After the intervention of antibiotics, in the antibiotic treatment group (FMT treatment group), except for *Proteobacteria*, most of the bacteria were killed. According to the Shannon, Simpson, and Chao1 index analyses, the abundance and diversity of bacteria in the mice treated with antibiotics decreased. This shows that relatively high concentrations of antibiotics can disrupt the original feed, high fats and sugars forming the structure of intestinal flora. The results of the Control group, Model group, and experiment where feces of mice treated with different doses of COST were transplanted in mice treated with antibiotics showed that fecal bacteria transplant inhibits abnormal serum lipid levels in mice. The accumulation of liver fat, the increase in fat, blood glucose, and glucose levels, and the abnormal function of insulin regulate the level of serum bile acid and have an obvious regulation effect on the structure of intestinal flora. The results show that COST can upregulate *Bacteroides*, *Lachnospiraceae*, *Lactobacillus,* and *Oscillospira* and downregulate *Desulfovibrio* bacteria through fecal bacteria transplantation. The results of heat map analysis showed that COST had a good regulation effect on *Clostridium*, *Dehalobacterium*, *Adlercreutzia*, *Akkermansia,* and *Streptococcus*. Combined with the above research results, we detected the genes related to lipid metabolism, and the results showed that the transplantation of bacterial flora also played a beneficial role in lipid metabolism. COST treatment and fecal bacteria transplantation after COST intervention play regulatory roles in *Akkermansia*, *Desulfovibrio,* and *Bacteroides*. We suspect that this group of microorganisms may be the core microbial community of lipid accumulation and lipid metabolism disorders treated by COST. In conclusion, COST can regulate the structure and diversity of intestinal flora. In addition, for a group of microorganisms, including *Bacteroides*, *Akkermansia,* and *Desulfovibrio,* the abundance and diversity of intestinal microorganisms after treatment continued to increase after direct COST intervention and fecal bacteria transplantation intervention. Therefore, this group of microorganisms may be potential core targets for COST to improve the lipid deposition and metabolic disorders caused by an HFHSD, and the specific functions of these microorganisms need to be further studied.

## 4. Materials and Methods

### 4.1. Animal Experimental Design

A total of 30 7-week-old male C57BL/6 mice were purchased from Hunan Slike Jingda Laboratory Animal Co., Ltd., No. 43004700064577 (Changsha, China), and raised in the Experimental Animal Center of Guangdong Pharmaceutical University. The animal center environment is Specific Pathogen Free (SPF) level; temperature: 24.0 ± 2.0 °C; relative humidity: 50–60%; air change times > 15 times/hour; light time: 12 h of alternating light and dark. After 1 week of adaptive feeding of common chow, all mice were randomly divided into 2 groups. CTRL group (*n* = 6) mice were given common chow (4.3% energy from fat, provided by Guangdong Provincial Medical Laboratory Animal Center, Guangzhou, China). Mice in the MOD group (*n* = 24) were given a high fat and high sugar diet (40% energy from fat; 17% energy from sucrose, Lot: D12327, Research Diets, Inc., New Brunswick, NJ, USA). Appendix A can be seen in feed formulation (Appendix A). After 8 weeks of feeding, the mice were randomly divided into 5 groups (6 mice in each group): CTRL group, MOD group, COST high-dose group (COST-H), COST medium-dose group (COST-M), and COST low-dose group (COST-L). According to the preliminary study of our research group, the dose of COST used in this experiment is as follows: COST-H, 1700 mg/kg/d; COST-M, 850 mg/kg/d, and COST-L, 425 mg/kg/d, the average Mw of chitosan oligosaccharides was ≤1000 Da (COST) (AK Biotech Co., Ltd. (Jinan, China)). The CTRL group mice were also given 0.9% NaCl solution for injection in equal amounts by intragastric administration, and their body weight was recorded weekly during the animal experiment. The physical and mental health of the mice was observed at each point in time when the data were recorded. After 12 weeks of intervention, the mice were anesthetized by inhalation of 1% pentobarbital sodium, and blood samples were collected through orbital blood collection. The collected blood, tissues, organs, and feces samples were frozen in liquid nitrogen and stored in a special refrigerator with ultra-low temperature biological samples for long-term storage.

Next, a pseudogerm-free mouse model was constructed. A total of 36 7-week-old male C57BL/6J mice were selected again. The mice were also purchased from Hunan Slike Jingda Laboratory Animal Co., Ltd., No. 110727201003661 (Changsha, China) and also raised in the Experimental Animal Center of Guangdong Pharmaceutical University.

After 1 week of adaptive feeding, all mice were randomly divided into 2 groups: pseudogerm-free mice (*n* = 30) and common chow (STD) mice (*n* = 6). The pseudogerm-free mice group was given mixed antibiotic drinking water and a high fat and high sugar diet (40% energy from fat; 17% energy from sucrose, Lot: D12327, Research Diets, Inc., New Brunswick, NJ, USA), Mice in the normal diet group were given ordinary sterile drinking water and common chow (4.3% energy from fat, provided by Guangdong Provincial Medical Laboratory Animal Center, Guangzhou, China). The antibiotic formula was mixed with ampicillin (1 g/L), neomycin sulfate (1 g/L), metronidazole (1 g/L), and vancomycin (0.5 g/L) in sterile water; the model establishment cycle is 4 weeks. During the period, the mice drank water and ate food freely, were weighed once a week, and the mental and activity status of the mice were monitored. After the model was established, the feces of the two groups of mice were collected for intestinal flora analysis to determine whether the model was successful.

After judging the success of modeling, the pseudogerm-free mice group was divided into 5 groups (6 mice in each group). The Control group, Model group, COST-H-FMT group, COST-M-FMT group, and COST-L-FMT group were given a high fat and high sugar diet, and the fecal flora extracted from the fresh fecal flora liquid was gavaged to pseudogerm-free mice at the rate of 100 μL/10 g/d for 6 weeks. The Control group was transplanted with fecal flora of the CTRL group, and the Model group was transplanted with fecal flora of the MOD group. Fecal flora of COST-H-FMT group mice was transplanted from donor mice in the COST-H group; Mice in the COST-M-FMT group were transplanted with feces from donor mice in the COST-M group. Mice in the COST-L-FMT group were transplanted with feces from donor mice in the COST-L administration group. At the same time, mice in the STD group were given the same amount of 0.9% NaCl solution for injection by intragastric administration and continued to receive common chow. During the fecal bacteria transplantation experiment, the body weight, mental state, and other physical indicators of the mice were observed and recorded. After 6 weeks of fecal bacteria transplantation intervention, mice were anesthetized with 1% pentobarbital sodium, the mice were dissected, and the blood was collected from the eye socket. The collected blood, tissues, organs, and feces samples were frozen in liquid nitrogen and stored in a special refrigerator with ultra-low temperature biological samples for long-term storage.

This experiment was approved by the Laboratory Animal Ethics Committee of Guangdong Pharmaceutical University, in strict accordance with the requirements of the ″Guidelines for the Ethical Review of Laboratory Animal Welfare” (GB/T35892-2018)to fully protect the welfare of laboratory animals.

### 4.2. Extraction Method of Fecal Flora Liquid

A total of 150–180 mg of feces was weighed and added to sterile 1mL phosphate buffer saline and sterile ceramic beads. A vortex oscillator was used to obtain suspension in an anaerobic environment. At 25 °C, centrifuged at 500 rpm for 10 min, the supernatant was quickly absorbed into another Eppendorf with a pipette. The supernatant was then centrifuged at 4000 rpm for 5 min, and the bacterial precipitate was retained. Sterile phosphate buffer saline was added again and washed twice more in the same way. Then, 60% sterile glycerol was added in the ratio of 1:2, mixed, and stored for a short time in an anaerobic environment.

### 4.3. Serum and Liver Index Analyses

Plasma levels of total triglyceride (TG), total cholesterol (TC), high-density lipoprotein cholesterol (HDL-C), low-density lipoprotein cholesterol (LDL-C), alanine aminotransferase (ALT), aspartate aminotransferase (AST), glucose, nonesterified fatty acids (NEFA), total bile acid (TBA), and liver levels of TC and TG were determined using various available kits (Nanjing Jiancheng Bioengineering Institute, Nanjing, China).

### 4.4. Histological Analyses

For the histological analysis, liver and adipose tissue samples were fixed in formalin solution for 24 h and then paraffin-embedded. Liver sections (4 mm thick) were stained with hematoxylin–eosin (HE). The stained samples were observed under a 200 × light microscope.

### 4.5. Faecal Microbiota 16S rRNA Analysis

16S rRNA (V3-V4 region) sequencing was used to analyze the composition of intestinal microbiota in feces. Metware Biotechnology (Wuhan, China) provided sequencing and analysis for the study. Specific experimental methods can be found in articles published by Li et al. [35].

Total DNA from feces was extracted according to the instructions of the QIAamp DNA Stool Mini Kit, DNA was quantified by Nanodrop, and 1.2% agarose was used for quality detection. According to the corresponding primers, the barcode sequence was added to perform PCR amplification on the rRNA gene variable region or specific gene fragment. The PCR amplification products were recovered, and the Quant-iT PicoGreen dsDNA Assay Kit reagent was used for fluorescence quantification on a microplate reader (FLx800, BioTek, Vermont, VT, USA) before sequencing. Sequencing library preparation was used to perform sequence end repair on the amplified products, A bases were added at the 3’ end of the DNA sequence, an index sequence was introduced at the 5’ end using BECKMAN AMPure XP Beads, and the library system was purified after adding adapters by magnetic beads. PCR amplification was conducted on the aforementioned DNA fragments connected to the adapter, the sequencing library template was enriched, BECKMAN AMPure XP Beads were used to purify the library again, and finally, 2% agarose gel electrophoresis was used to further purify the library. Libraries were quality-checked on an Agilent Bioanalyzer, quantified on a Promega QuantiFluor Fluorescence Quantitation System using Quant-iT PicoGreen dsDNA Assay Kit reagents, and then sequenced onboard.

### 4.6. Quantitative RT–PCR

The relevant gene sequences of mouse and human origin were searched on NCBI, and Shanghai Biotech Co., Ltd. (Shanghai, China) was commissioned to design and synthesize the upstream and downstream primers. The primer sequences are shown in Appendix A. Total RNA was isolated from mouse liver tissue using TRIzol reagent. Reverse transcription of RNA was performed according to the manual provided with the PrimeScript™ RT Reagent Kit with gDNA Eraser (Perfect Real Time), and the acquired cDNA was frozen at −20 °C until use. The synthesized cDNA was subjected to fluorescence quantitative PCR (FQ-PCR) using the TaKaRa SYBR Premix EX TaqTM kit in a reaction system volume of 10 μL. The reaction program was as follows: predenaturation at 95 °C for 30 s; 40 cycles of denaturation at 95 °C for 5 s, annealing at 60 °C for 20 s and extension at 72 °C for 20 s; and data collection at 65 °C for 15 s and melting curve analysis as the temperature increased from 60 °C to 95 °C at a rate of 0.5 °C every 5 s. After the reaction was complete, the data were processed using the 2^−ΔΔCt^ method for the calculation of relative gene expression.

### 4.7. Glucose Tolerance Test (OGTT)

The mice were fasted for 12 h, during which they could freely drink water. Mice were gavaged with glucose at a dose of 1.5 g/kg. The blood glucose values at 0 min, 15 min, 30 min, 60 min, and 120 min were measured sequentially with a Roche blood glucose meter. After the experiment, the dietary conditions set in the experimental protocol were restored.

## 5. Conclusions

In this study, COST was shown to improve diet-induced lipid metabolism disorder for the first time, and the target of the effect was suspected to be the intestinal microflora treated by intestinal microbe COST. Then, by establishing a pseudogerm-free mice mouse model, the fecal flora of mice treated with transplanted COST was used to explore the effect of COST intervention on the intestinal flora. The results of this study showed that COST could improve intestinal microecology and regulate lipid metabolism disorder in vivo to achieve a lipid-lowering effect. In particular, COST may play a central regulatory role in microbiota, including *Bacteroides*, *Akkermansia,* and *Desulfovibrio*. These findings indicate that COST is a very effective natural marine product with activity against lipid metabolism disorders.

## Figures and Tables

**Figure 1 marinedrugs-20-00444-f001:**
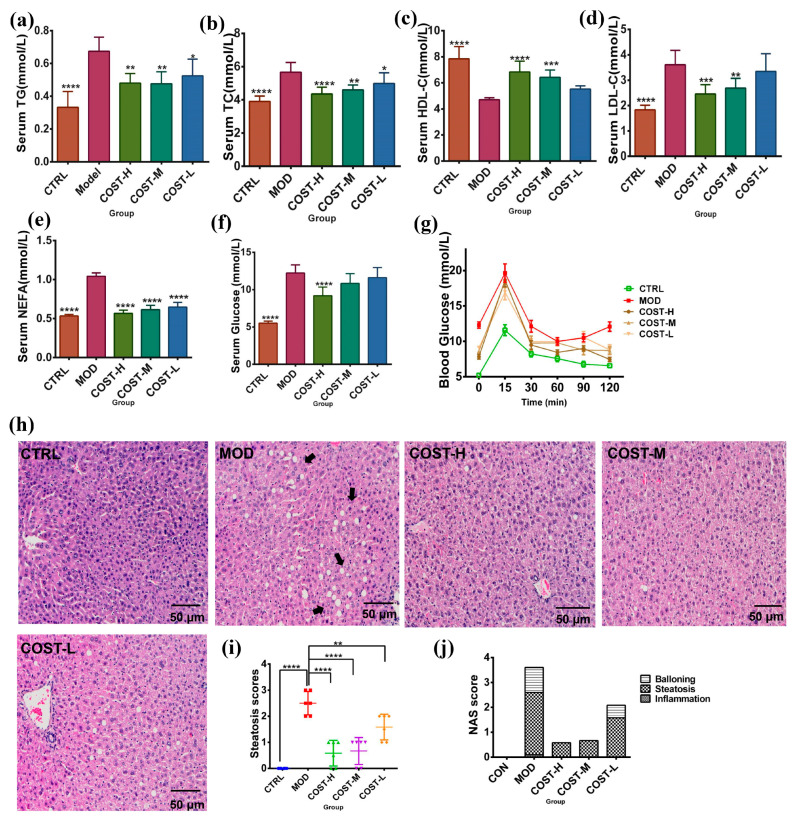
Changes of serum biochemical indexes and glucose tolerance in mice after COST treatment and observation of liver pathological lesions. (**a**) Serum TG level (**b**) Serum TC level (**c**) Serum HDL-C level (**d**) Serum LDL-C level (**e**) Serum NEFA level (**f**) Serum glucose level (**g**) Glucose tolerance (**h**) pathological sections of liver, the arrows point to the distribution of lipid droplets in the liver. (**i**,**j**) liver histopathology score. Data are expressed as the mean ± SEM, *n* = 6, * *p* < 0.05, ** *p* < 0.01, *** *p* < 0.001, **** *p* < 0.0001 vs. the MOD group.

**Figure 2 marinedrugs-20-00444-f002:**
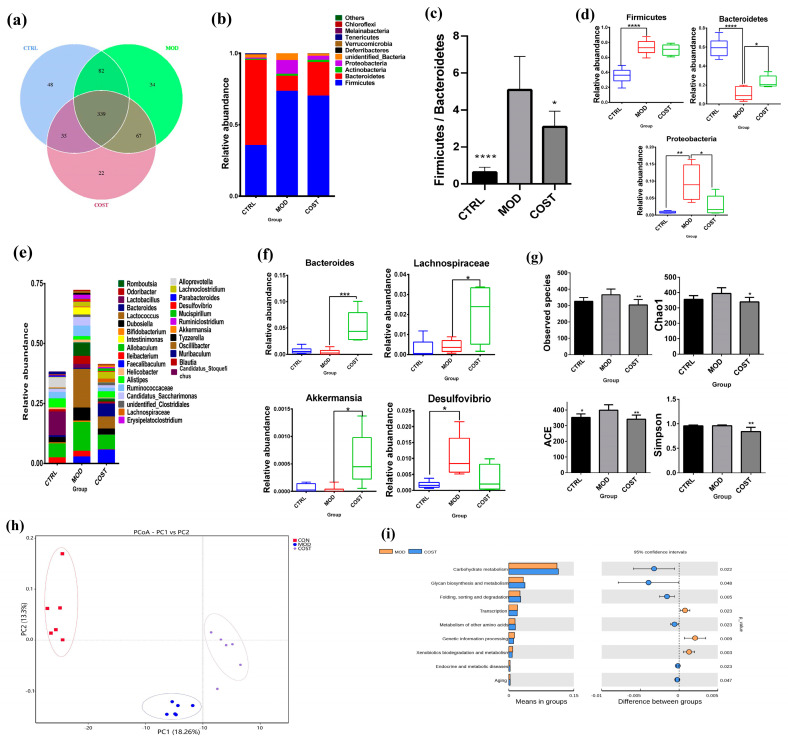
Changes in gut microbial structure after COST treatment. (**a**) OTUs level (**b**) phylum-level species composition differences. (**c**) F/B value (**d**) Marker species at phylum level (**e**) Genus-level species composition differences (**f**) Marker species at genus level (**g**) Alpha diversity analysis (**h**) PCoA analysis based on Unweighted Unifrac distance (**i**) KEGG pathway level analysis. Data are expressed as the mean ± SEM, *n* = 6, * *p* < 0.05, ** *p* < 0.01, *** *p* < 0.001, **** *p* < 0.0001 vs. the MOD group.

**Figure 3 marinedrugs-20-00444-f003:**
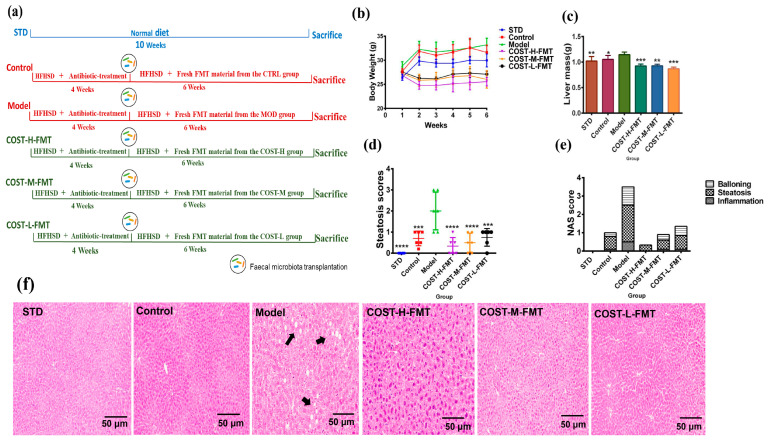
Differences in body weight, liver weight, adipose tissue weight, and pathological sections of liver and perirenal adipose tissue in mice treated with fecal bacteria transplantation. (**a**) FMT experiment summary (**b**) Weekly changes in body weight of mice (**c**) Liver weight level (**d**,**e**) liver histopathology score (**f**) pathological sections of liver, the arrows point to the distribution of lipid droplets in the liver. Data are expressed as the mean ± SEM, *n* = 6, * *p* < 0.05, ** *p* < 0.01, *** *p* < 0.001, **** *p* < 0.0001 vs. the Model group.

**Figure 4 marinedrugs-20-00444-f004:**
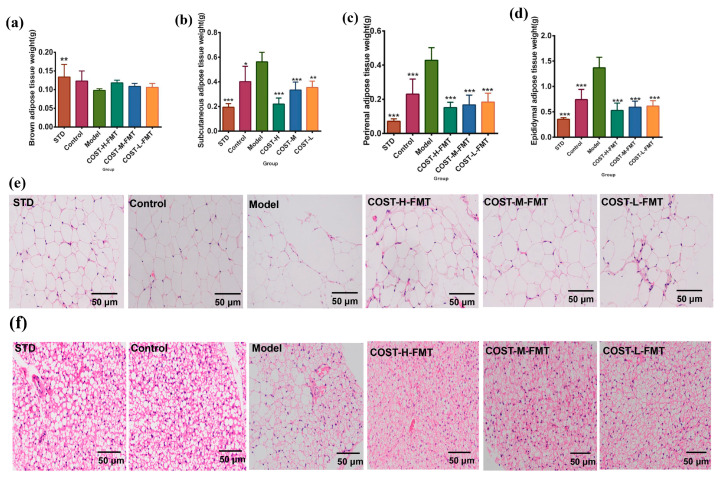
Differences in adipose tissue weight and pathological sections of brown adipose tissue and perirenal adipose tissue in mice treated with fecal bacteria transplantation. (**a**) Brown adipose tissue weight level (**b**) Subcutaneous adipose tissue weight level (**c**) Perirenal adipose tissue weight level (**d**) Epididymal adipose tissue weight (**e**) Pathological sections of perirenal adipose tissue (**f**) Pathological sections of brown adipose tissue. Data are expressed as the mean ± SEM, *n* = 6, * *p* < 0.05, ** *p* < 0.01, *** *p* < 0.001 vs. the Model group.

**Figure 5 marinedrugs-20-00444-f005:**
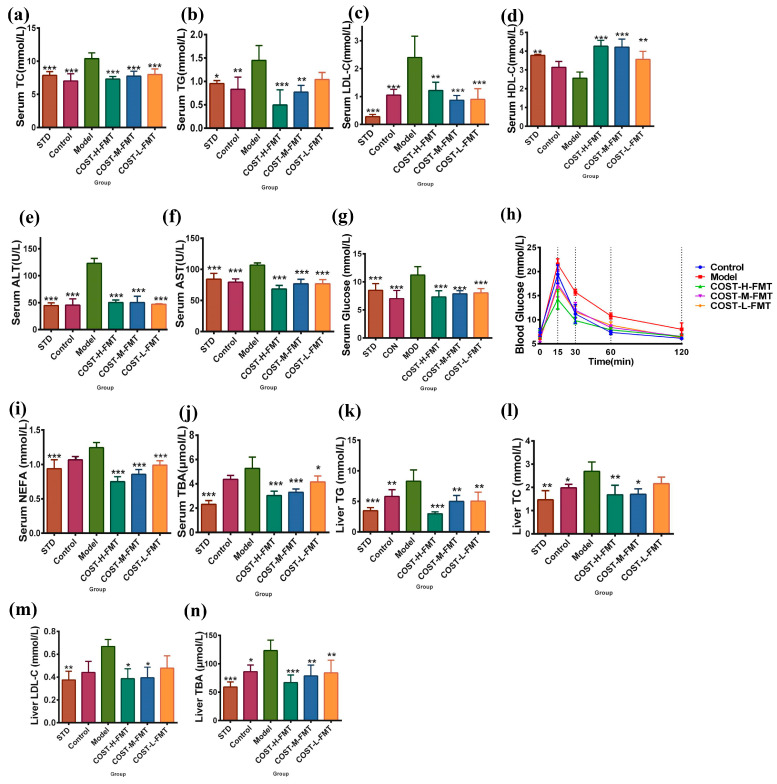
Changes in serum and liver biochemical levels and glucose tolerance in mice treated with FMT. (**a**) Serum TC level (**b**) Serum TG level (**c**) Serum LDL- C level (**d**) Serum HDL level (**e**) Serum ALT level (**f**) Serum AST level (**g**) Serum glucose level (**h**) Glucose tolerance (**i**) Serum NEFA level (**j**) Serum TBA level (**k**) Liver TG level (**l**) Liver TC level (**m**) Liver LDL- C level (**n**) Liver TBA level. Data are expressed as the mean ± SEM, *n* = 6, * *p* < 0.05, ** *p* < 0.01, *** *p* < 0.001 vs. the Model group.

**Figure 6 marinedrugs-20-00444-f006:**
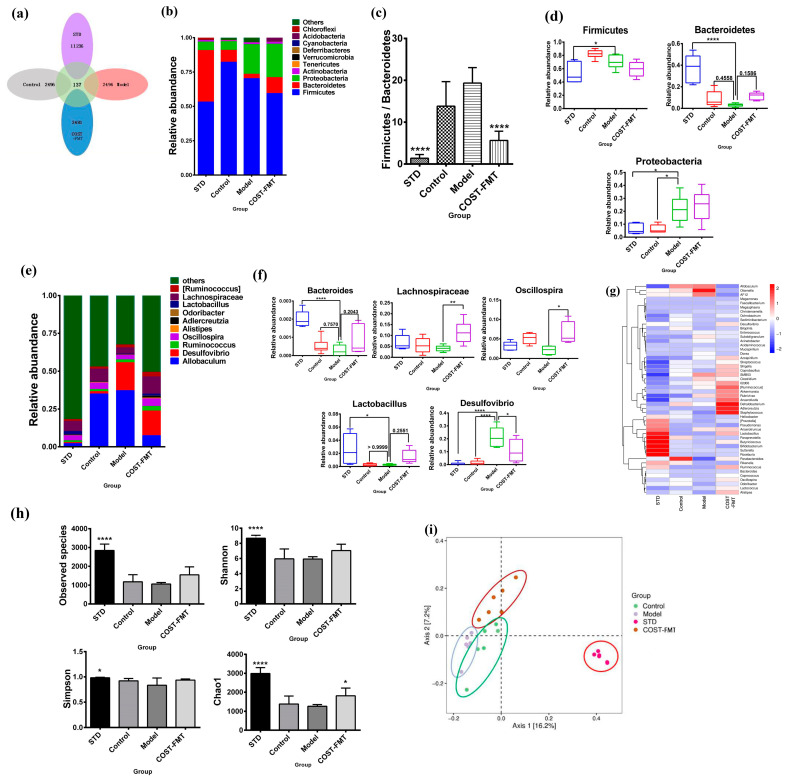
Changes in gut microbial structure in mice treated with fecal microbiota transplantation. (**a**) OTUs level. (**b**) Phylum-level species composition differences. (**c**) F/B value. (**d**) Marker species at phylum level. (**e**) Genus-level species composition differences. (**f**) Marker species at genus level. (**g**) Heat map analysis of species composition at genus level. (**h**) Alpha diversity analysis. (**i**) PCoA analysis based on Unweighted Unifrac distance. Data are expressed as the mean ± SEM, *n* = 6, * *p* < 0.05, ** *p* < 0.01, **** *p* < 0.0001 vs. the Model group.

**Figure 7 marinedrugs-20-00444-f007:**
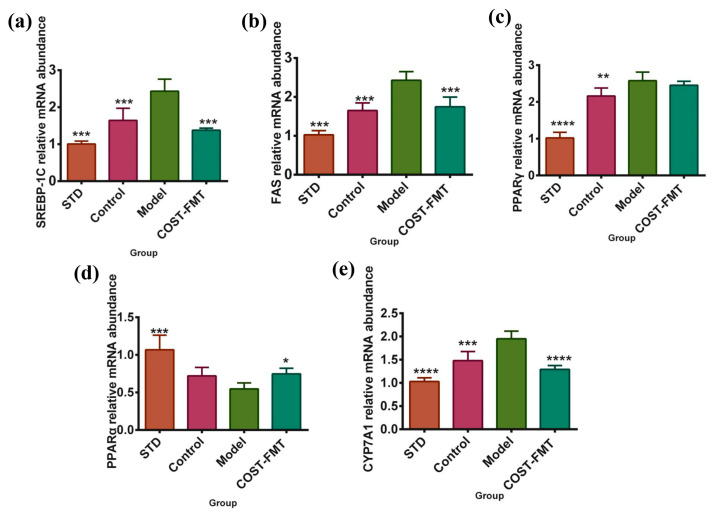
The expression of lipid metabolism related genes in FMT experimental mice (**a**) SREBP-1C mRNA expression. (**b**) FAS mRNA expression. (**c**) PPARγ mRNA expression. (**d**) PPARα mRNA expression. (**e**) CYP7A1 mRNA expression. Data are expressed as the mean ± SEM, *n* = 6, * *p* < 0.05, ** *p* < 0.01, *** *p* < 0.001, **** *p* < 0.0001 vs. the Model group.

## Data Availability

The data presented in this study are available within the article.

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
