# Peer review of "The Ameliorative Effect of COST on Diet-Induced Lipid Metabolism Disorders by Regulating Intestinal Microbiota"

_marinedrugs, 2022, doi:10.3390/md20070444_

Round 1
Reviewer 1 Report
The manuscript by Huimin You et al analyzes the effects of chito-oligosaccharides (COST), in mice with metabolic alterations produced by a diet high in fat and sugar and its relationship with the intestinal microbiota.
Different biochemical and histological parameters, the effects on the microbiota and fecal transplant findings are included. Although the effects seem significant, it is necessary a thoroughly revision of the manuscript and a presentation improvement:
Main comments:
1. 4.1 Animal experimental design (page 12, line 428): it is necessary a full revision; check the pseudogerm-free mouse model description (from 30 mice, two groups, pseudogerm-free mice (n = 30) and normal chow SDT (n=6) were established; describe treatments uniformly in weeks (4 weeks) or in days (28 days); page 13, line 460: …” the mice drank water and food freely and were weighed once a week to monitor the mental and activity status …”
2. Page 13, line 464: The extracted fresh fecal flora liquid; in page 12, line 446, it is mentioned that “ After 8 weeks, fresh feces was collected from each group of mice, and approximately 300 mg of feces was collected from each mouse and stored at -80°C”. A better description of samples used for fecal transplant should be included.
3. Page 2, line 88; “The serum lipoprotein cholesterol (HDL-C) levels in the COST-H and COST-M groups were similar to those in the CTRL group and showed an upward trend compared with the MOD group (Figure 1a-e; Table S1)”. The description if for Figure 1c only; p<0.0001 and p<0.001 appear in the figure).
4. Changes are described in the CTRL group, when in fact these mice have no manipulation (page 3, line 132: “ Compared with the MOD group, the abundance of Firmicutes and Proteobacteria 132 in the CTRL group was significantly decreased (P < 0.001, P < 0.01), while the abundance of Bacteroidetes was increased (P < 0.001)”.
5. Page 5, line 189: The description of brown and white adipose tissue should be in a separate paragraph Data in Tables 1 and 2 are mean SD and in the figures are mean SEM
6. Figures: The font size used (numbers and letters) in the figures is very small and even with zoom it is difficult to read
7. Pictures with pathological sections of liver (Figs 1 and 3) should include arrows, arrowheads and other symbols to point the alterations described in the text.
8. Many sentences have repeated terms (page 3, line 114; page 5, line 169; page 5, line 174; page 13, line 459).
Author Response
Point-by-Point Response Letter (Manuscript Number: marinedrugs-1782769)
Dear editors and reviewers,
Thank you for your letter and the reviewers’ comments regarding our manuscript titled “The ameliorative effect of COST on diet-induced lipid metabolism disorders by regulating intestinal microbiota” (Manuscript Number: marinedrugs-1782769). These comments were very helpful in revising and improving our paper. We studied these comments carefully and made corresponding corrections that we hope will meet with your approval. The changes in the revised manuscript have been marked in red in the supplementary material. The responses to the editor and reviewers’ comments are provided below.
Response to Editor’s comments and Reviewers:
Thank you for the time and effort spent in reviewing the manuscript. These suggestions and comments are very helpful to us.
Responses to Reviewer #1 comments:
Thank you for taking the time and energy to review the manuscript. These suggestions and opinions are very helpful to us.
Comments:
- 4.1 Animal experimental design (page 12, line 428): it is necessary a full revision; check the pseudogerm-free mouse model description (from 30 mice, two groups, pseudogerm-free mice (n = 30) and normal chow SDT (n=6) were established; describe treatments uniformly in weeks (4 weeks) or in days (28 days); page 13, line 460: …” the mice drank water and food freely and were weighed once a week to monitor the mental and activity status”
Response to comment:
Thank you very much for your constructive comments. We correct this part as follows:
Thirty 7-week-old male C57BL/6 mice were purchased from Hunan Slike Jingda Laboratory Animal Co., Ltd., No. 43004700064577(Hunan, China), and raised in Ex-perimental Animal Center of Guangdong Pharmaceutical University. The animal cen-ter environment is Specific Pathogen Free(SPF) level, temperature: 24.0±2.0°C, relative humidity: 50-60%, air change times >15 times/hour, light time: 12 hours of alternating light and dark. After 1 week of adaptive feeding of common chow, all mice were ran-domly divided into 2 groups: CTRL group (n = 6) mice were given common chow (4.3% energy from fat, provided by Guangdong Provincial Medical Laboratory Animal Cen-ter) , Mice in MOD group (n = 24) were given a high-fat and high-sugar diet (40% energy from fat; 17% energy from sucrose, Lot: D12327, Research Diets, Inc. U.S.). Supplementary material can be seen in feed formulation (Table S3). After 8 weeks of feeding, the mice were randomly divided into 5 groups (6 mice in each group): CTRL group, MOD group, COST high-dose group (COST-H), COST medium-dose group (COST-M) and COST low-dose group (COST-L). According to the preliminary study of our research group, the dose of COST used in this experiment is as follows: COST-H,1700 mg/kg/dï¼›COST-M,850 mg/kg/d and COST-L,425 mg/kg/d,the average Mw of chitosan oligosaccharides was ≤1000 Da (COST) (AK Biotech Co., Ltd. (Shandong China)),the CTRL group mice were also given 0.9% NaCl solution for injection in equal amount by intragastric administration, and their body weight was recorded weekly during the animal experiment. The physical and mental health of the mice was observed at each point in time when the data were recorded. After 12 weeks of intervention, the mice were anesthetized by inhalation of 1% pentobarbital sodium and blood samples were collected through orbital blood collection. The collected blood, tis-sues, organs and faeces samples are frozen in liquid nitrogen, and stored in a special refrigerator with ultra low temperature biological samples for long-term storage.
Next, pseudogerm-free mouse model was constructed: thirty-six 7-week-old male C57BL/6J mice were selected again. The mice were also purchased from Hunan Slike Jingda Laboratory Animal Co., Ltd., No. 110727201003661 (Hunan, China) and also raised in Experimental Animal Center of Guangdong Pharmaceutical University.
After 1 week of adaptive feeding, all mice were randomly divided into two groups: pseudogerm-free mice (n = 30) and common chow (STD) mice (n = 6). The pseudogerm-free mice group was given mixed antibiotic drinking water and a high-fat and high-sugar diet (40% energy from fat; 17% energy from sucrose, Lot: D12327, Research Diets, Inc. U.S.),Mice in the normal diet group were given ordinary sterile drinking water and common chow (4.3% energy from fat, provided by Guangdong Provincial Medical Laboratory Animal Center). The antibiotic formula is mixed with ampicillin (1 g/L), neomycin sulfate (1 g/L), metronidazole (1 g/L) and vancomycin (0.5 g/L) in sterile water, the model establishment cycle is four weeks. During the period, the mice drank water and food freely, were weighed once a week, and the mental and activity status of the mice were monitored. After the model was established, the feces of the two groups of mice were collected for intestinal flora analysis to determine whether the model was successful.
After judging the success of modeling, the pseudogerm-free mice group was divided into 5 groups (6 mice in each group) : Control group, Model group, COST-H-FMT group, COST-M-FMT group and COST-L-FMT group were given a high-fat and high-sugar diet, and the fecal flora extracted from the fresh fecal flora liquid was gavaged to pseudogerm-free mice at the rate of 100 μL/10 g/d for 6 weeks. Control group was transplanted with fecal flora of CTRL group, Model group was transplanted with fecal flora of MOD group. Fecal flora of COST-H-FMT group mice was transplanted from donor mice in COST-H group; Mice in COST-M-FMT group were transplanted with feces from donor mice in COST-M group. Mice in COST-L-FMT group were transplanted with feces from donor mice in COST-L administration group. At the same time, mice in STD group were given the same amount of 0.9% NaCl solution for injec-tion by intragastric administration and continued to receive common chow. During the fecal bacteria transplantation experiment, the body weight, mental state and other physical indicators of the mice were observed and recorded. After 6 weeks of fecal bacteria transplantation intervention, mice were anestheized with 1% pentobarbital sodium, and the mice were dissected and the blood was collected from the eye socket, the collected blood, tissues, organs and faeces samples are frozen in liquid nitrogen, and stored in a special refrigerator with ultra low temperature biological samples for long-term storage.
This experiment was approved by the Laboratory Animal Ethics Committee of Guangdong Pharmaceutical University, in strict accordance with the requirements of the "Guidelines for the Ethical Review of Laboratory Animal Welfare (GB/T35892-2018) "to fully protect the welfare of laboratory animals.
- Page 13, line 464: The extracted fresh fecal flora liquid; in page 12, line 446, it is mentioned that “ After 8 weeks, fresh feces was collected from each group of mice, and approximately 300 mg of feces was collected from each mouse and stored at -80°C”. A better description of samples used for fecal transplant should be included.
Response to comment:
Thank you for your thoughtful advice, We added this section in 4.2 as follows:
4.2 Extraction method of fecal flora liquid
Weighing 150-180 mg fecal, add sterile 1mL phosphate buffer saline and sterile ceramic beads, and use vortex oscillator to get suspension in anaerobic environment. At 25℃, centrifuged at 500 rpm for 10 min, the supernatant was quickly absorbed into another eppendorf with a pipette. The supernatant was then centrifuged at 4000 rpm for 5 min, and the bacterial precipitate was retained. Add sterile phosphate buffer sa-line again and wash twice more in the same way. 60% sterile glycerol was added in the ratio of 1:2, and then mixed, and stored for a short time in an anaerobic environment.
3.Page 2, line 88; “The serum lipoprotein cholesterol (HDL-C) levels in the COST-H and COST-M groups were similar to those in the CTRL group and showed an upward trend compared with the MOD group (Figure 1a-e; Table S1)”. The description if for Figure 1c only; p<0.0001 and p<0.001 appear in the figure).
Response to comment:
We apologize for our negligence, we modify the content of this section as follows:The plasma levels of TG, TC and LDL-C decreased in the high-dose and medium-dose COST groups (P < 0.01) compared with the MOD group (Figure 1a-b, Figure 1d; Table S1). NEFA levels in the high-dose, medium-dose and low-dose COST groups were significantly lower than those in the MOD group (P < 0.0001) (Figure 1e; Table S1). The serum lipoprotein cholesterol (HDL-C) levels in the COST-H and COST-M groups were similar to those in the CTRL group and showed an upward trend compared with the MOD group (Figure 1c; Table S1).
Figure 1. Changes of serum biochemical indexes and glucose tolerance in mice after COST treatment and observation of liver pathological lesions. (a) Serum TG level (b) Serum TC level (c) Serum HDL- C level (d) Serum LDL-C level (e) Serum NEFA level (f) Serum glucose level (g) Glucose tolerance (h) pathological sections of liver. (i-j) liver histopathology score. Data are expressed as the mean±SEM, n = 6, * P < 0.05,** P < 0.01,*** P < 0.001, **** P < 0.0001 vs. the MOD group.
- Changes are described in the CTRL group, when in fact these mice have no manipulation (page 3, line 132: “Compared with the MOD group, the abundance of Firmicutes and Proteobacteria 132 in the CTRL group was significantly decreased (P < 0.001, P < 0.01), while the abundance of Bacteroidetes was increased (P< 0.001)”.
Response to comment:
We are sorry to have troubled you because of our negligence, We correct this section as follows: Compared with the MOD group, the abundance of Firmicutes and Proteobacteria in the CTRL group was significantly decreased (P < 0.001, P < 0.01), while the abundance of Bacteroidetes was increased (P < 0.001). Compared with MOD group, the COST group was able to upregulate the abundance of Bacteroidetes (P < 0.05) and downregulate the abundance of Proteobacteria (P < 0.05)( Figure 2d).
Figure 2. Changes in gut microbial structure after COST treatment. (a) OTUs level (b) phylum-level species composition differences. (c) F/B value (d) Marker species at phylum level (e) Genus-level species composition differences (f) Marker species at genus level (g) Alpha diversity analysis (h) PCoA analysis based on Unweighted Unifrac distance (i) KEGG pathway level analysis. Data are expressed as the mean±SEM, n = 6, * P < 0.05,** P < 0.01,*** P < 0.001, **** P < 0.0001 vs. the MOD group.
- Page 5, line 189: The description of brown and white adipose tissue should be in a separate paragraph Data in Tables 1 and 2 are mean SD and in the figures are mean SEM
Response to comment:
- Thank you for your valuable comments, we split this part into two parts(Lines 173-176)
Effects of FMT on body weight and liver in mice
Next, the fecal bacteria transplantation experiment was carried out on pseudogerm-free mice that were successfully modeled. The specific experimental method is shown in the Figure 3a. During the FMT experiment, the weekly body weight of C57BL/6J mice in each group was recorded (Figure 3b, Table S2). Macroscopically, compared with other FMT groups, the Model group gained more weight, and there were significant differences between the Model group and the other four groups. The fecal transplanted from the MOD group mice with lipid metabolism disorder can promote lipid metabolism disorder. The results of the COST-H-FMT, COST-M-FMT and COST-L-FMT groups showed that the high-dose group, in particular, could main-tain the initial body weight of mice well, and could effectively resist the effect of pro-moting lipid metabolism disorder caused by high-fat and high-sugar diet compared with the Model group (P < 0.001), and all could inhibit the metabolic disorders caused by the HFHSD diet. No significant difference was found in the final body weight of FMT mice treated with three doses of COST, therefore, no significant dose-dependent effect of COST transplantation intervention was found. The liver weight of the Model group was compared (Figure 3c, Table S2), and the liver weight of the Model group was significantly higher than that the COST-H-FMT, COST-M-FMT and COST-L-FMT group (P < 0.05) Next, our observation of pathological tissue sections in each group showed (Figure 3f) no obvious liver lipid deposition in the STD, Control, COST-H-FMT, COST-M-FMT and COST-L-FMT groups. However, liver lipid accumulation in the Model group was obvious. And After transplantation of feces of mice receiving COST intervention (Figure 3d-e), NAS score was also significantly reduced, especially for steatosis score(P < 0.001).These results suggest that transplantation of feces from donor mice with metabolic disorders can increase the liver weight and the risk of liver steatosis in recipient mice, and transplantation of feces of mice receiving COST intervention can effectively improve this situation.
Figure 3. Differences in body weight, liver weight, adipose tissue weight, and pathological sections of liver and perirenal adipose tissue in mice treated with fecal bacteria transplantation. (a) FMT experiment summary (b) Weekly changes in body weight of mice (c) Liver weight level (d-e) liver histopathology score (f) pathological sections of liver. Data are expressed as the mean±SEM, n = 6, * P < 0.05,** P < 0.01,*** P < 0.001, **** P < 0.0001 vs. the Model group.
Effects of FMT and adipose tissue in mice
The main function of white adipose tissue (WAT) is to store excess fat in the body, which is a common mechanism and manifestation of lipid metabolism. The main func-tion of brown adipose tissue (BAT) is to break down lipid molecules and convert them into CO2, H2O and heat energy, which helps increase the body's metabolism and WAT consumption and is converted into beige fat cells [23, 24]. Therefore, we compared the adipose tissues of mice in each group. For BAT (Figure 4a, Table S2), the content of BAT in the STD group was the highest, while that in the Model group was the lowest, but the difference was not significant. On the other hand, WAT analysis of different parts (Figure 4b-d, Table S2) showed that fecal bacteria transplantation of different donor groups in the FMT experiment had a considerable impact on the WAT of mice. The WAT content of the standard diet and transplanted intestinal flora in the STD and Control groups was significantly lower than that in the Model group (P <0.05). Similarly, the WAT contents in the COST-H-FMT COST-M-FMT and COST-L-FMT trans-plantation groups were significantly lower than those in the MOD transplantation group (P < 0.001). This result indicated that the intestinal flora of mice fed a normal diet and mice after COST intervention could effectively resist the accumulation of WAT. Next, we also observed pathological sections of perirenal adipose tissue(Figure 4e) and brown adipose tissue (Figure 4f) and observed that the fat cells of the STD group and Control group had normal structures, orderly arrangements, small cell volumes and a large number of fat cells in the same field of vision. In contrast, the WAT cells in the Model group were uneven in size, with a small number of cells and more obvious cell proliferation and amplification. The sections of the COST-H-FMT and COST-M-FMT groups were similar to those of the STD and Control groups. Compared with the Model group, the adipocytes were signifi-cantly smaller with uniform size, smaller volume and better lesions, but COST-L-FMT showed no significant improvement effect. These results indicate that the treatment of intestinal flora with COST can indirectly inhibit the fat cell hypertrophy caused by HFHSD, regulate the growth and expansion of fat cells, reduce fat accumulation, and improve the symptoms of abnormal lipid metabolism.
Figure 4 . Differences in boadipose tissue weight, and pathological sections of brown adipose tissue and perirenal adipose tissue in mice treated with fecal bacteria transplantation. (a) Brown adipose tissue weight level (b) Subcutaneous adipose tissue weight level (c) Perirenal adipose tissue weight level (d) Epididymal adipose tissue weight (e) Pathological sections of perirenal adipose tissue (f) Pathological sections of brown adipose tissue. Data are expressed as the mean±SEM, n = 6, * P < 0.05,** P < 0.01,*** P < 0.001 vs. the Model group.
- I'm sorry for our mistakes,in Table S1 and S2 should be mean SEM, we have corrected that
- Figures: The font size used (numbers and letters) in the figures is very small and even with zoom it is difficult to read.
Response to comment:
Thank you for your thoughtful advice, we are very sorry about it. We have rearranged the layout and size of the image, as follows:
Figure 1. Changes of serum biochemical indexes and glucose tolerance in mice after COST treatment and observation of liver pathological lesions. (a) Serum TG level (b) Serum TC level (c) Serum HDL- C level (d) Serum LDL-C level (e) Serum NEFA level (f) Serum glucose level (g) Glucose tolerance (h) pathological sections of liver. (i-j) liver histopathology score. Data are expressed as the mean±SEM, n = 6, * P < 0.05,** P < 0.01,*** P < 0.001, **** P < 0.0001 vs. the MOD group.
Figure 2. Changes in gut microbial structure after COST treatment. (a) OTUs level (b) phylum-level species composition differences. (c) F/B value (d) Marker species at phylum level (e) Genus-level species composition differences (f) Marker species at genus level (g) Alpha diversity analysis (h) PCoA analysis based on Unweighted Unifrac distance (i) KEGG pathway level analysis. Data are expressed as the mean±SEM, n = 6, * P < 0.05,** P < 0.01,*** P < 0.001, **** P < 0.0001 vs. the MOD group.
Figure 3. Differences in body weight, liver weight, adipose tissue weight, and pathological sections of liver and perirenal adipose tissue in mice treated with fecal bacteria transplantation. (a) FMT experiment summary (b) Weekly changes in body weight of mice (c) Liver weight level (d-e) liver histopathology score (f) pathological sections of liver. Data are expressed as the mean±SEM, n = 6, * P < 0.05,** P < 0.01,*** P < 0.001, **** P < 0.0001 vs. the Model group.
Figure 4 . Differences in boadipose tissue weight, and pathological sections of brown adipose tissue and perirenal adipose tissue in mice treated with fecal bacteria transplantation. (a) Brown adipose tissue weight level (b) Subcutaneous adipose tissue weight level (c) Perirenal adipose tissue weight level (d) Epididymal adipose tissue weight (e) Pathological sections of perirenal adipose tissue (f) Pathological sections of brown adipose tissue. Data are expressed as the mean±SEM, n = 6, * P < 0.05,** P < 0.01,*** P < 0.001 vs. the Model group.
Figure 5. Changes in serum and liver biochemical levels and glucose tolerance in mice treated with FMT. (a) Serum TC level (b) Serum TG level (c) Serum LDL- C level (d) Serum HDL level (e) Serum ALT level (f) Serum AST level (g) Serum glucose level (h) Glucose tolerance (i) Serum NEFA level (j) Serum TBA level (k) Liver TG level (l) Liver TC level (m) Liver LDL- C level (n) Liver TBA level. Data are expressed as the mean±SEM, n = 6, * P < 0.05,** P < 0.01,*** P < 0.001 vs. the Model group.
Figure 6. Changes in gut microbial structure in mice treated with fecal microbiota transplantation. (a) OTUs level. (b) Phylum-level species composition differences. (c) F/B value. (d) Marker species at phylum level (e) Genus-level species composition differences. (f) Marker species at genus level.(g) Heat map analysis of species composition at genus level. (h) Alpha diversity analysis. (i) PCoA analysis based on Unweighted Unifrac distance. Data are expressed as the mean±SEM, n = 6, * P < 0.05,** P < 0.01,*** P < 0.001, **** P < 0.0001 vs. the Model group.
Figure 7. The expression of lipid metabolism related genes in FMT experimental mice (a) SREBP-1C mRNA expression. (b) FAS mRNA expression. (c) PPARγ mRNA expression. (d) PPARα mRNA expression. (e) CYP7A1 mRNA expression. Data are expressed as the mean±SEM, n = 6, * P < 0.05,** P < 0.01,*** P < 0.0011, **** P < 0.0001 vs. the Model group.
- Pictures with pathological sections of liver (Figs 1 and 3) should include arrows, arrowheads and other symbols to point the alterations described in the text.
Response to comment:
Thank you for your meaningful comments, in Figures 1 and 3, we indicate the distribution of lipid droplets in the liver with arrows to better illustrate the variation in the text
- Many sentences have repeated terms (page 3, line 114; page 5, line 169; page 5, line 174; page 13, line 459).
Response to comment:
I'm sorry for the inappropriate use of my language,We have commissioned American Journal Experts for further careful checking of manuscripts and have made careful corrections and editing,and the following corrections were made to the error
- According to the above activity results, COST can reduce lipid accumulation and has the potential to resist lipid metabolism disorders, and one of the key features of lipid metabolism disorders is the influence of gut microbes.
- Next, the fecal bacteria transplantation experiment was carried out on the pseudogerm-free mice that were successfully modeled. The specific experimental method is shown in the figure.
- Macroscopically, compared with other FMT groups, the Model group gained more weight, and there were significant differences between the Model group and the other four groups. The fecal transplanted from the MOD group mice with lipid metabolism disorder can promote lipid metabolism disorder. The results of the three COST-FMT groups showed that the high-dose group, in particular, could maintain the initial body weight of mice well, and could effectively resist the effect of promoting lipid metabolism disorder caused by high-fat and high-sugar diet compared with the Model group (P < 0.001).
- …, the model establishment cycle is four weeks. During the period, the mice drank water and food freely, were weighed once a week, and the mental and activity status of the mice were monitored.

Reviewer 2 Report
This paper by Huimin You et al describes that Chitosan oligosaccharides (COST) can change intestinal microbiota and ameliorate lipid metabolism disorder induced by high fat and high sugar diet. Microbiota composition of each group is well analyzed and the biochemical levels are shown cost dose dependent manner. So, these data are interesting and sufficiently written, but several points should be addressed by the authors.
<Major comments>
1. The author suggested that COST can change murine intestinal microbiome with several analysis in Figure 2. However, diet you adopted here are provided from different companies, causing an uncertainty microbiota composition of “CTRL” may different with that of “MOD” regardless of COST treatment. Please provide a clear evidence about the similarity of fecal microbiota composition between mice fed with “Guangdong Provincial Medical Laboratory Animal Center” and “Research Diets (Cat #. D12329)”.
2. According to the data, the body weight and adipose tissue of all COST-treated group was decreased in dose-dependent manner. It is suspected that these phenotypes may be derived from the toxicity of COST itself. So, please check the toxicity of this material on mouse to confirm that the COST did not effect on food uptake of mice.
<Minor review>
1. Please provide liver histopathology score and scale bar on Fig. 1h, 3h and 3i.
2. Please modify a grammatical expression and add appropriate references on P3L115.
Author Response
Point-by-Point Response Letter (Manuscript Number: marinedrugs-1782769)
Dear editors and reviewers,
Thank you for your letter and the reviewers’ comments regarding our manuscript titled “The ameliorative effect of COST on diet-induced lipid metabolism disorders by regulating intestinal microbiota” (Manuscript Number: marinedrugs-1782769). These comments were very helpful in revising and improving our paper. We studied these comments carefully and made corresponding corrections that we hope will meet with your approval. The changes in the revised manuscript have been marked in red in the supplementary material. The responses to the editor and reviewers’ comments are provided below.
Response to Editor’s comments and Reviewers:
Thank you for the time and effort spent in reviewing the manuscript. These suggestions and comments are very helpful to us.
Responses to Reviewer #2 comments:
Thank you for taking the time and energy to review the manuscript. These suggestions and opinions are very helpful to us.
<Major comments>
- The author suggested that COST can change murine intestinal microbiome with several analysis in Figure 2. However, diet you adopted here are provided from different companies, causing an uncertainty microbiota composition of “CTRL” may different with that of “MOD” regardless of COST treatment. Please provide a clear evidence about the similarity of fecal microbiota composition between mice fed with “Guangdong Provincial Medical Laboratory Animal Center” and “Research Diets (Cat #. D12329)”.
Response to comment:
Thank you for your constructive suggestion. We conduct species classification tree statistics on the kingdom, phylum, class, order, family, genus, and species of gut microbes in the CTRL group and the MOD group, so as to compare the similarity of species distribution between the two groups, it can be seen from the results in the figure that at the phylum level, Fimicutes, Bacteroidetes, and Actinobactrria in the CTRL group and the MOD group are widespread, common, representative and shared microbial species in the distribution of gut microbes,We then conducted a similar analysis at the class, order, family, genus, and species level of these three representative species. At the class level, both the CTRL group and the MOD group shared the three species undefined_Actinobactrria, Bacteroidia, and Bacilli. Then, at the order level, the CTRL group and the MOD group share the four species Bifidobacteriales, Bacteroidales, Lactobacillales and Erysipelotrichales. At the family level, the CTRL group and the MOD group share the Bifidobacteriaceae, Bacteroidaceae, Marinifilaceae, Lactobacillaceae, Streptococcaceae, Erysipelotrichaceae, At the genus level, Bifidobacterium, Bacteroides, Odoribacter, Lactobacillus, Lactococcus, Allobaculum, Dubosiella, Erysipelatoclostridium, Faecalibaculum, lleibacterium were shared by both CTRL and MOD groups. At the species level, Bifidobacterium_animalis, Bacteroides_acidifaciens, Lactobacillus_reuteri, Lactococcus_lactis, Faecalibaculum_rodentium, lleibacterium_valens were shared by both CTRL and MOD groups, This indicated that although the gut microbial species composition of the CTRL and MOD groups had certain similarities.
Figure. Analysis of gut microbes in CTRL group(a) and MOD group(b) based on taxtree
- According to the data, the body weight and adipose tissue of all COST-treated group was decreased in dose-dependent manner. It is suspected that these phenotypes may be derived from the toxicity of COST itself. So, please check the toxicity of this material on mouse to confirm that the COST did not effect on food uptake of mice.
Response to comment:
According to our previous research results, according to the "Technical Specifications for Inspection and Evaluation of Health Foods", acute toxicity test, systematic genotoxicity test, Ames test, bone marrow cell micronucleus test and 30-day feeding test were carried out. Among them, the results of the 30-day feeding test showed that during the test period, the male and female rats in each dose group of COST had good growth, no abnormal symptoms and signs, and no death; Compared with the control group, there was no significant difference in food intake, food intake and food utilization rate; the values ​​of blood routine indexes and blood biochemical indexes were all within the normal physiological range; there was no significant difference in the ratio of main organs and organs compared with the control group; no histopathological observation was found. Any abnormal changes, the results show that the above results show that the maximum unobserved adverse effect dose of COST on rats is above 5.00 g/kg·bw, which is more than 100 times the recommended human dose of 0.05 g/kg·bw, which further confirms that COST is a safe and non-toxic substances. It is confirmed that COST is a safe and non-toxic substance. Reference[1] detailed research results can be found as follows:
<Minor comments>
- Please provide liver histopathology score and scale bar on Fig. 1h, 3h and 3i.
Thank you very much for your constructive comments, we are sincerely sorry. We have added liver histopathology score and scale bar to the figures
Figure 1. Changes of serum biochemical indexes and glucose tolerance in mice after COST treatment and observation of liver pathological lesions. (a) Serum TG level (b) Serum TC level (c) Serum HDL- C level (d) Serum LDL-C level (e) Serum NEFA level (f) Serum glucose level (g) Glucose tolerance (h) pathological sections of liver. (i-j) liver histopathology score. Data are expressed as the mean±SEM, n = 6, * P < 0.05,** P < 0.01,*** P < 0.001, **** P < 0.0001 vs. the MOD group.
Figure 3. Differences in body weight, liver weight, adipose tissue weight, and pathological sections of liver and perirenal adipose tissue in mice treated with fecal bacteria transplantation. (a) FMT experiment summary (b) Weekly changes in body weight of mice (c) Liver weight level (d-e) liver histopathology score (f) pathological sections of liver. Data are expressed as the mean±SEM, n = 6, * P < 0.05,** P < 0.01,*** P < 0.001, **** P < 0.0001 vs. the Model group.
Figure 4 . Differences in boadipose tissue weight, and pathological sections of brown adipose tissue and perirenal adipose tissue in mice treated with fecal bacteria transplantation. (a) Brown adipose tissue weight level (b) Subcutaneous adipose tissue weight level (c) Perirenal adipose tissue weight level (d) Epididymal adipose tissue weight (e) Pathological sections of perirenal adipose tissue (f) Pathological sections of brown adipose tissue. Data are expressed as the mean±SEM, n = 6, * P < 0.05,** P < 0.01,*** P < 0.001 vs. the Model group.
- Please modify a grammatical expression and add appropriate references on P3L115.
1.Thank you for your valuable comments and suggestions. we apologize for grammar issues. We have commissioned American Journal Experts for further careful spelling and grammar checking of manuscripts and have made careful corrections and editing.
- Thank you for your constructive advice, we have added references after this sentence as a basis, as follows: (Lines 118-120)
According to the above activity results, COST can reduce lipid accumulation and has the potential to resist lipid metabolism disorders, and one of the key features of lipid metabolism disorders is the influence of gut microbes[2-5].
- Ye Zhijun. Toxicological evaluation of cost safety and the effect of costg on lipid metabolism in obese rats. Guangdong Pharmaceutical University (2019).doi: 10.27690/d.cnki.ggdyk.2019.000065.
- Z. Wang; Koonen D.; Hofker M.; Fu J.: Gut Microbiome and Lipid Metabolism: From Associations to Mechanisms. Curr Opin Lipidol 2016, 27, 216-24.
- A.A. Kolodziejczyk; Zheng D.; Shibolet O.; Elinav E.: The Role of the Microbiome in Nafld and Nash. EMBO Mol Med 2019, 11,
- C.W. Ko; Qu J.; Black D.D.; Tso P.: Regulation of Intestinal Lipid Metabolism: Current Concepts and Relevance to Disease. Nat Rev Gastroenterol Hepatol 2020, 17, 169-83.
- M. Nieuwdorp; Gilijamse P.W.; Pai N.; Kaplan L.M.: Role of the Microbiome in Energy Regulation and Metabolism. Gastroenterology 2014, 146, 1525-33.

Round 2
Reviewer 1 Report
This reviewer appreciates the effort made by the authors to incorporate the suggestions in the manuscript. The resulting version is much more complete and easier to understand.
Author Response
Dear editors and reviewers,
Thank you for your letter and the reviewers’ comments regarding our manuscript titled “The ameliorative effect of COST on diet-induced lipid metabolism disorders by regulating intestinal microbiota” (Manuscript Number: marinedrugs-1782769).
Thank you very much for reading my reply letter. I have learned a lot from you. Your suggestions are of great guiding significance for my thesis writing and scientific research work.
Kind regards,
Zhengquan Su
Reviewer 2 Report
To author and editor:
I carefully check the response letter based on your data, but you did not answer my question and requirements exactly.
The data you attached in major comment #1 is shown well in terms of distribution of gut microbiome between CTRL and MOD group. However, the percentage of each components is extremely changed which is key trigger to induce diverse human diseases, containing metabolic disorder.
So, you have to present the microbiota sequencing data containing the similarity of fecal microbiota composition between mice fed with “Guangdong Provincial Medical Laboratory Animal Center” and “Research Diets (Cat #. D12329)”.
Author Response
Dear editors and reviewers,
Thank you for your letter and the reviewers’ comments regarding our manuscript titled “The ameliorative effect of COST on diet-induced lipid metabolism disorders by regulating intestinal microbiota” (Manuscript Number: marinedrugs-1782769). I am very sorry that I could not answer your questions and requirements due to my inaccurate reply. In this regard, we have once again carried on the following supplement
First of all, thank you very much for your careful reading of our reply letter. As mentioned before, according to Taxtree analysis (Figure 1), CTRL and MOD have roughly the same species composition structure of intestinal microbes at the four levels of kingdom, phylum, class, order, family, genus and species. But, as you mentioned, the percentage of each component has changed dramatically, which is a key trigger for a variety of human diseases, including metabolic disorders. The main reason for this is that the mice in the lipid metabolism disorder group, namely the MOD group, were fed a high-fat and high-sugar diet (FAT accounted for 20.4%, Carbohydrate accounted for 46.1%, Protein accounted for 23%, Fiber accounted for 5.5%), The control group we established was CTRL group, and CTRL group were fed with normal diet (FAT accounted for 4.3%, Carbohydrate accounted for 0%, Protein accounted for 20%, Fiber accounted for 4.8%), The CTRL mice were healthy, normal mice without disease, their gut microbes were undamaged by an unhealthy diet high in fat and sugar, our study confirms what the researchers have stated[1-3], The high-fat and high-sugar diet caused great changes in the percentage of each component in the intestinal microbe composition of mice with such lipid metabolism disorder. For example, especially at the phylum level, the characteristic flora of Fimicutes and Bacteroidetes was greatly changed. By establishing these two groups, it was proved that COST could reverse the intestinal microbiome disorder caused by high fat and high sugar diet compared with MOD group, and the intestinal microbiome could be restored to the normal level of CTRL group, the raw data of gut microbiome partial sequencing are shown at the end.
Figure1. Analysis of gut microbes in CTRL group(a) and MOD group(b) based on taxtree.
References
- J. Yin; Li Y.; Han H.; Chen S.; Gao J.; Liu G.; Wu X.; Deng J.; Yu Q.; Huang X.; Fang R.; Li T.; Reiter R.J.; Zhang D.; Zhu C.; Zhu G.; Ren W.; Yin Y.: Melatonin Reprogramming of Gut Microbiota Improves Lipid Dysmetabolism in High-Fat Diet-Fed Mice. J Pineal Res 2018, 65, e12524.
- A. Walker; Pfitzner B.; Neschen S.; Kahle M.; Harir M.; Lucio M.; Moritz F.; Tziotis D.; Witting M.; Rothballer M.; Engel M.; Schmid M.; Endesfelder D.; Klingenspor M.; Rattei T.; Castell W.Z.; de Angelis M.H.; Hartmann A.; Schmitt-Kopplin P.: Distinct Signatures of Host-Microbial Meta-Metabolome and Gut Microbiome in Two C57bl/6 Strains under High-Fat Diet. ISME J 2014, 8, 2380-96.
- G. Jamar; Ribeiro D.A.; Pisani L.P.: High-Fat or High-Sugar Diets as Trigger Inflammation in the Microbiota-Gut-Brain Axis. Crit Rev Food Sci Nutr 2021, 61, 836-54.
We would like to express our great appreciation to you and the reviewers for the comments regarding our paper. If you have any further questions, please do not hesitate to contact us.
Kind regards,
Zhengquan Su

Round 3
Reviewer 2 Report
Thank you for your answer exactly that I want.
Thank you again for allowing us to review this interesting work, and I look forward to evaluation future research from your group.